# Environmental and Social Factors Associated with High Chronic Kidney Disease Mortality Rates in Municipalities of Guatemala: An Ecological Study of Municipal-Level Mortality Data

**DOI:** 10.3390/ijerph20085532

**Published:** 2023-04-17

**Authors:** Alejandro Cerón

**Affiliations:** Department of Anthropology, University of Denver, Denver, CO 80208, USA; Alejandro.CeronValdes@du.edu

**Keywords:** chronic kidney disease, chronic kidney disease of non-traditional causes, environmental indicators, Guatemala

## Abstract

The purpose of this study was to determine the association between social and environmental indicators and high mortality rates from chronic kidney disease (CKD) in municipalities of Guatemala. An ecological study of municipal-level factors associated with CKD mortality in Guatemala was conducted. Crude mortality rates were calculated for the 2009–2019 period for each of the country’s 340 municipalities, by gender and age groups. Municipal-level social and environmental indicators were used as independent variables. Linear regression was used for bivariate and multivariate analysis. A total of 28,723 deaths from CKD were documented for the 2009–2019 period. Average crude mortality rate for all ages for the country’s 340 municipalities was 70.66 per 100,000 [0–502.99]. Very highly positive associations with high mortality rates were found in two agrarian territories where land use is mainly for permanent crops (e.g., sugar cane, coffee, rubber, banana, plantain, African palm) and pastures for cattle, with very low percentages of land covered by forests or protected areas. Social factors related to poverty and environmental factors related to agricultural use of land may play a role in the high CKD mortality rates documented in a cluster of municipalities of Guatemala.

## 1. Introduction

The incidence, prevalence, and mortality of chronic kidney disease (CKD) has increased globally in the past four decades [1]. The worldwide rise in CKD cases is attributable to the increased number of people living with diabetes and high blood pressure, with other less common causes such as infection, injury, exposure to toxins, or environmental pollution [1]. A type of CKD that is not related to diabetes and hypertension has been described as CKD of non-traditional causes (CKDnT) [2,3,4,5,6,7]. Although this paper focuses on mortality from CKD in general, it remains that CKDnT has reached epidemic proportions in Central America, making it a public health priority [4], and that studies that focus on CKDnT may shed some light on the present study’s findings. Most researchers attribute CKDnT to multifactorial causes, with an interplay of repeated heat exposure and poor hydration working in concert with an exposure to toxins and infectious agents to damage the kidneys [2,3,4,5,6]. While environmental factors such as exposure to agrochemicals or heavy metals have been hypothesized to play a role in CKDnT, available evidence is inconclusive [4,5,6,7,8,9,10,11,12,13], and research on this area is insufficient in Latin American countries [10].

Efforts at using environmental health indicators to influence public health decision-making have increased since the 1990s [14]. Initial attempts at adding environmental and health indicators [14,15,16,17,18,19,20,21,22,23] have evolved into the implementation of environmental public health tracking programs in some middle- or high-income countries [24,25,26,27,28,29,30]. Additionally, environmental science specialists have developed a “socio ecological system” approach to emphasize the need to look at the interactions between social, economic, and ecological phenomena [31]. In Guatemala, environmental scholars have produced empirical data that utilizes the socio-ecological system approach at the municipal level to generate a typology of ten agrarian zones [32,33].

Chronic kidney disease is a public health priority in Guatemala. It is one of the diseases that contributes most to the global burden of disease, measured as years of life lost due to disability (DALYs), and it has a profound impact on the quality of life of many populations, as well as on the resources of health systems [34]. Guatemala is one of the countries in which CKD has a greater impact on DALYs and mortality [34,35], with an age-standardized CKD mortality rate estimated at 47.8 deaths per 100,000 people, only exceeded in the region of the Americas by Nicaragua, Mexico, and El Salvador, with age-standardized mortality estimates of 49.4, 58.1, and 71.4 deaths per 100,000 people, respectively [34]. A different source estimated Guatemala’s age-standardized mortality rate from kidney diseases at 55.1, only below Bolivia’s 55.8, El Salvador’s 72.9, and Nicaragua’s 73.9 [35]. These estimates are compatible with a previous study among selected Central American countries [36]. According to data from the Department of Epidemiology of the Ministry of Public Health and Social Assistance, the prevalence and mortality from CKD have increased significantly since 2008, showing greater increases in warm regions, with low schooling rates and high rates of low-skill jobs [37]. The few studies available suggest that diabetes and high blood pressure are the main factors associated with CKD in the general population, although some also suggest that there are population segments in which occupational or environmental exposures to the agricultural industry or heavy metals may play a role [38,39,40,41,42,43]. The implementation of public health actions for the prevention and control of CKD requires special efforts to address the risk factors and social determinants that make certain social groups particularly vulnerable to this disease [44,45,46]. In addition to health promotion and disease prevention measures, timely diagnosis and access to comprehensive treatment play a key role in improving people’s quality of life and reducing the impact of the complications of the disease on society and the health system [47].

Guatemala has a highly segmented and fragmented healthcare system, composed of a network of public, private nonprofit, and private for-profit institutions [48]. With an underfunded public sector, private healthcare facilities have proliferated in the past two decades, with a meager availability of health care insurance and out-of-pocket expenditure accounting for more than half of the country’s total health expenditure, resulting in deep disparities in health care access and utilization [49]. Causes of mortality show an almost even split between cardiovascular diseases, accidents and violence, communicable diseases, and tumors [50,51], with inequities affecting indigenous people, those living in rural areas, and those without formal education [52,53]. Geographic analyses of mortality distribution show high levels of inequalities between municipalities, [50,51,54,55,56,57]. Previous studies of CKD in Guatemala have shown a spatial distribution of prevalence [12,13] and mortality [37], but an analysis of the social and environmental determinants of these differences is non-existent.

The goal of this study was to determine the association between social and environmental indicators and high mortality rates from CKD in municipalities of Guatemala. The study had the following aims: (1) to calculate municipal mortality rates for CKD for the general population and selected age- and sex-specific groupings relevant to CKDnT debates; (2) to determine the association between municipal mortality rates and municipalities’ agrarian zone classification; and (3) to determine the association between municipal mortality rates and other social or environmental indicators relevant to CKDnT debates.

## 2. Materials and Methods

This was an ecological study of municipal-level social and environmental factors associated with chronic kidney disease (CKD) mortality in Guatemala from 2009–2019 using a cross-sectional (i.e., not longitudinal) analysis with vital registration data.

### 2.1. Data

The crude mortality rates for each municipality were calculated using, for the numerator, Guatemala’s official annual deaths information for the years 2009 to 2019 [58] using the operational definition for “chronic kidney disease” utilized by the GBD 2019 Diseases and Injuries Collaborators [1], which includes the following International Classification of Diseases, Tenth Revision (ICD-10) codes: D63.1, E08.2-E08.29, E10.2-E10.29, E11.2-E11.29, E12.2, E13.2-E13.29, E14.2, I12-I13.9, N02-N08.8, N15.0, N17-N19, Q60-Q63.2, Q63.8-Q63.9, Q64.2-Q64.9, Z49-Z49.32, Z52.4, Z99.2. Municipality of residence, as opposed to municipality where the death occurred, was used for the analysis. Municipalities’ populations from Guatemala’s 2018 census were used for the denominators because there were no available reliable estimates for municipal-level populations for the years 2009–2019 [59]. Crude mortality rates were calculated for the 2009–2019 period for each of the country’s 340 municipalities’ total population of all ages, male population of all ages, female population of all ages, population under 60 years, population under 45 years, and population under 20 years. The sex- and age-specific groupings respond to the Pan American Health Organization’s (PAHO) chronic kidney disease of non-traditional causes (CKDnT) case definitions—which recommend one to consider deaths of CKD among people under 60 years as potentially due to CKDnT [60]—and to literature suggesting that deaths from CKDnT may occur in pediatric populations or in young adults [37,43,44]. Guatemala has not implemented a CKDnT-specific ICD-10 code, as it has been recommended by PAHO [60].

For independent variables, the Institute of Agriculture, Natural Resources and Environment of the Rafael Landívar University’s (IARNA) classification of municipalities into ten agrarian territories based on their agricultural and rurality situations [61] (shown in Figure 1) was used. Finally, municipality-level data on average temperature, altitude [62], percentage of people dedicated to agriculture, area used for agriculture [63], percentage of people living in poverty [33], number of health care facilities [64], and percentage of Mestizo/Ladino population [59] were included as independent variables, given their relevance to discussions about CKDnT and its relation to the social and environmental indicators.

### 2.2. Statistical Analysis

A database (Appendix A) was built with the mortality rates and the socio environmental indicators by municipality. All agrarian territories were included with dummy variables as potential predictors, as were other relevant variables at the municipal level: average temperature, altitude, percentage of people dedicated to agriculture, area used for agriculture, percentage of people living in poverty, number of health care facilities, and percentage of Mestizo/Ladino population. Statistical analysis used STATA^®^ 17 (StataCorp. 2021. Stata Statistical Software: Release 17. College Station, StataCorp LLC, College Station, TX, USA). For each of the five age- or sex-specific mortality rates, descriptive statistics were estimated. Linear regression was used for bivariate and multivariate analysis, using an alpha of 0.05 to test if associations were statistically significant.

## 3. Results

A total of 28,723 deaths from chronic kidney disease (CKD) were documented for the period between 2009 and 2019, with 53% corresponding to males. The number of CKD deaths increased by 80% (82% for males, 78% for females) in this period. Table 1 shows the distribution of CKD deaths by age and ethnicity for males, females, and the general population. The majority of CKD deaths (61%) occurred among those sixty years and older, while 6% occurred among those under 20 years of age, with similar age distributions for both sexes. For ethnicity, 45% of the total CKD deaths were among the Mestizo/Ladino, and 16% were among the Maya, with similar distributions for both sexes.

The average crude mortality rate for all ages for the country’s 340 municipalities was 70.66 per 100,000, with a range of 0 to 502.99. Table 2 shows descriptive statistics for the municipal crude CKD mortality rates for the entire population and for each of the age- and sex-specific mortality rates. Comparing male and female mortality rates, while the average and maximum rates are much higher among males, the median, first and third quartiles are similar for both groups, showing that there are municipalities in the highest quartile where the difference by sex is important. Comparing the three different age groups shows increasingly higher mortality rates in the groups that include older populations.

The geographic distribution of mortality rates by municipality suggests a pattern in which the highest rates tend to concentrate in the South and Southwest agrarian regions, especially for the population of all ages, male population, under-60 population, and under-45 population. While the female population and the under 20 population also show their highest mortality rates in those regions, they also show high rates scattered across the country. Figure 2 shows this geographic distribution for the total population and for each of the age and sex groups, with the darkest shaded municipalities representing the highest mortality rates.

Looking at the municipalities with the highest CKD mortality rates in the general population, and in the sex- and age-specific groups, there are twelve municipalities included in at least five of the six top-25 lists (Table 3), all of which are located in the South and Southwest agrarian regions.

Figure 3 shows the location of the twelve municipalities on a map of Guatemala. The six municipalities included in the top 25 for all six sex- and age-specific groups are San José La Máquina, Ocós, Cuyotenango, Tiquisate, Ayutla, and Champerico, and they tend to be located in the Southwest agrarian region. The six municipalities included in the top 25 for five of the six top 25 lists did not make the top 25 for the under-20 population, and they tended to be located in the South agrarian region. Those municipalities are Retalhuleu, San José El Ídolo, San Miguel Panán, La Gomera, Moyuta, and Chiquimulilla.

Bivariate analysis showed very strong positive and negative associations for some of the Institute of Agriculture, Natural Resources and Environment of the Rafael Landívar University’s (IARNA) agrarian territories (Table 4). For the under-20 population, the only significant association was with the “5. East” agrarian territory (regression coefficient 5.31, *p* < 0.05). For the other five groups, there were very highly significant positive associations (*p* < 0.01) with the agrarian territories “4. Concentrated in the south and dispersed in the highlands” (regression coefficients between 23.06 for under 45 population, and 47.37 for male population) and “7. Southwest” (regression coefficients between 22.20 for female population, and 81.16 for male population); there were very highly significant negative associations (*p* < 0.01) with the agrarian territory “2. North transversal strip” (regression coefficients between −21.11 in under 45 population and −52.28 in male population) and with the agrarian territory “1. Northwest highlands” (regression coefficients between −11.98 in under 45 population and −36.79 in male population); and highly significant negative associations (*p* < 0.05) with the agrarian territory “8. Petén” (regression coefficients between −22.78 in under 45 population and −56.09 in male population).

Table 5 shows the results of a bivariate analysis for the other relevant variables, showing very highly significant associations for average temperature, altitude, poverty, and percentage of Mayan (indigenous) or Mestizo/Ladino (non- indigenous) populations for all age-and sex-specific groups, with the exception of the under 20 population. Associations were not significant for percentage of agriculture workforce, percentage of agricultural area, and available health care facilities.

The multivariate analysis included the two agrarian regions that showed significant positive associations with mortality rates in the bivariate analysis (i.e., southwest and south), and it included the other relevant variables that showed a significant association, positive or negative (i.e., average temperature, altitude, poverty, and Mestizo/Ladino population). The Mayan (indigenous) population was not included in the multivariate analysis due to it being highly correlated with the Mestizo/Ladino (non-indigenous) population, with both representing mirror images of each other and with Maya presenting a negative association and Mestizo/Ladino a positive one. Table 6 shows the results of the multivariate analysis, where the two agrarian regions and poverty keep similar coefficients to the ones they showed in the bivariate analysis and with very high significance, while the temperature, altitude, and Mestizo/Ladino variables lost significance.

## 4. Discussion

This study found very highly significant associations between municipal chronic kidney disease (CKD) mortality rates and several agrarian territories and poverty levels for all the analyzed gender and age groups. Associations with the other social and environmental factors involved in this study, including temperature, altitude, and ethnicity, did not show statistical significance in the multivariate analysis. The lack of association with temperature, altitude, and ethnicity should not be interpreted as a lack of relevance of these variables in the etiology of CKD, but as directly related to the ecologic nature of this analysis, especially since these variables interact dynamically with other variables at the individual level of analysis, which was outside of the scope of the present study. The highest coefficients by far were those associated with agrarian territories. The geographic distribution found in this study supports findings reported by Laux et al. [13], based on dialysis enrollment, and by Sam Colop [37], based on vital statistics; however, this study adds the multivariate analysis associations to social and environmental factors and to gender and age groups, pointing to an overlap of traditional and non-traditional exposures in a cluster of municipalities.

Very highly positive associations with CKD mortality rates were found in territories “4. Concentrated in the south and dispersed in the highlands,” and “7. Southwest,” which included 41 and 51 municipalities, respectively. Municipalities in these two territories showed moderate human development indicators, the majority being the Ladino/Mestizo (non-indigenous) population and the rural population, with agriculture being the main economic activity. Land use in these territories is mainly for permanent crops (e.g., sugar cane, coffee, rubber, banana, plantain, African palm) and pastures for cattle, with a lower percentage of land used for temporary crops (e.g., corn, black beans, sesame) and a very low percentages of land covered by forests or protected areas. There is a concentration of land in large plantations. Land erosion levels are high [61]. In contrast, very highly negative associations with CKD mortality rates were found in the territories “1. Northwest highlands,” “2. North transversal strip,” and “8. Petén.” Although these three territories vary among themselves, the main difference with territories 4 and 7 is that the majority of their land is used for temporary crops, forests, or protected areas [61]. Municipal poverty level was also positively associated, with a very high significance. Notably, availability of healthcare facilities, percentage of people working in agriculture, land area devoted to agriculture, average temperature, altitude, and percentage of Ladino/Mestizo population did not show significant associations in the multivariate analysis.

Given that there is a consensus among researcher regarding chronic kidney disease of non-traditional causes (CKDnT) and the role of agricultural work and heat exposure [2,3,4,5,6], this study’s results should be interpreted in light of the following. A high proportion of agricultural workers in these areas, especially those working in sugar cane plantations, have their place of residence in neighboring or distant municipalities [37,42,45]; therefore, the analysis presented here, which is based on place of residence, was not able to capture those migratory patterns. Similarly, available information did not allow us to analyze in detail the deceased’s type of occupation, but the majority performed what is classified as “elementary occupations” in the current International Standard Classification of Occupations, ISCO-08 [37]. Regarding availability of healthcare services, specialized kidney diagnosis and treatment were not captured in the analyzed data, given that those specialized services, including access to dialysis, tend to be centralized in a few municipalities [11,38]. Additional research is needed to understand the life cycle development of the disease among male and female populations in the affected communities, including their occupational histories, as well as whether there may be a connection between exposure at the worksite and contamination in the household or community of residence.

Of the top 25 municipalities with the highest CKD mortality rates for different sex- or age-specific groups (Table 3, Figure 3, and Appendix A), the number of municipalities that belong to territories 4 or 7 is 21 (84%) in the population of all ages, 22 (88%) in the male population, 17 (68%) in the female population, 21 (84%) in the under-60 population, 16 (64%) in the under-45 population, and 8 (32%) in the under 20 population [61]. Moreover, six municipalities (San José La Máquina, Ocós, Cuyotenango, Tiquisate, Ayutla, and Champerico) were among the top 25 for the six population groups examined, and six more (Retalhuleu, San José El Ídolo, San Miguel Panán, La Gomera, Moyuta, and Chiquimulilla) were in the top 25 for all but one of the groups, namely the under-20 group.

In terms of the public health and epidemiological investigation of traditional and novel or emerging risk factors and determinants of CKD, this study’s findings seem to support hypotheses that point to environmental factors related to agricultural use of land and to poverty. Although it is reasonable to interpret from these findings that there are traditional risk factors such as diabetes and hypertension driving most of the CKD deaths, it also seems plausible to identify a cluster of municipalities with very high mortality rates across different age- and sex-specific groups, wherein some type of environmental exposure associated with agricultural use of land and/or with poverty is also at play in tandem with traditional risk factors of CKD. Given that exposure to agrochemicals or heavy metals has been hypothesized to play a role in CKDnT, but that available evidence is inconclusive [4,5,6,7,8,9,10,11,12,13], the environmental study of the identified cluster of municipalities offers a promising way forward for testing environmental hypotheses. Similarly, it is known that poverty is associated to traditional risk factors such as diabetes and hypertension, but it may be associated with other factors that affect people at younger ages, such as nutrition, as some studies have found [65], or to other cardiometabolic exposures.

The limitations of this study are that the mortality data came from national vital statistics, and the independent variables came from secondary data generated by the Institute of Agriculture, Natural Resources and Environment of the Rafael Landívar University (IARNA) for a different purpose. The municipality-level analysis in which this study is based, by definition, is susceptible to the insertion of the ecological fallacy into the analysis. It was outside the scope of this manuscript to differentiate traditional CKD mortality from CKDnT mortality. Another limitation is that, given that there are no available reliable population estimates by municipality for the intercensal years between 2002 and 2018, this analysis used the 2018 census data for the denominators; the effect of this limitation likely does not impact the general conclusions of this analysis, but it could have an effect on the mortality rates for specific municipalities that have had dramatic changes in fertility, mortality, or migration rates, but the demographic information that would allow for the identification of those municipalities is not yet available. The strengths of this study are that it is one of the few population-based ecologic studies analyzing environmental indicators in Central America [8,12], and that it is compatible with the Pan American Health Organization’s CKDnT research priorities, protocols and operational definitions [44,60]. This study also looked at age group and gender differences, which have rarely been investigated in this line of research. It is important to take into account that the analyses that come from vital statistics databases anywhere in the world have limitations in the quality of the definition of the variables and in the consistency with which they are interpreted by those who generate the data. The strength of these data is that they cover the entire population and are generated periodically. It is worth considering that the quality of the data generated by the Guatemalan vital statistics system has recently been assessed as good [66] and as “four stars” out of five [1].

## 5. Conclusions

The results from this study suggest that social factors related to poverty and environmental factors related to agricultural use of land play a role in the high CKD mortality rates documented in a cluster of municipalities of Guatemala.

Future research aimed at investigating social and environmental exposures in high CKD mortality municipalities is needed, in addition to population-based studies using a life-cycle approach, including gender differences and the role of occupational trajectories. Ideally, an interdisciplinary, multi-level cohort study that collects data at the individual and community levels and that also collects social and environmental data would help get a more comprehensive view of the drivers of increased mortality rates in these municipalities. Public health interventions aimed at the early detection of decreasing renal function in high-CKD-mortality municipalities are recommended. Public health interventions should begin with active surveillance using screening tests from individuals’ renal function, as well as screening tests from potential community sources of pollutants.

## Figures and Tables

**Figure 1 ijerph-20-05532-f001:**
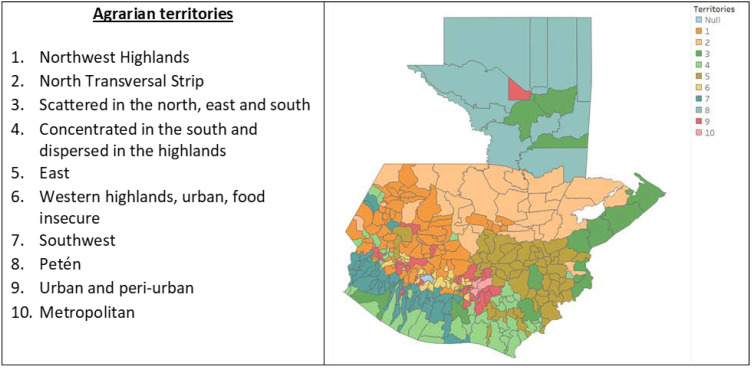
Agrarian territories and municipalities of Guatemala. Created by author, based on Instituto de Agricultura, Recursos Naturales y Ambiente de la Universidad Rafael Landívar (IARNA) [61].

**Figure 2 ijerph-20-05532-f002:**
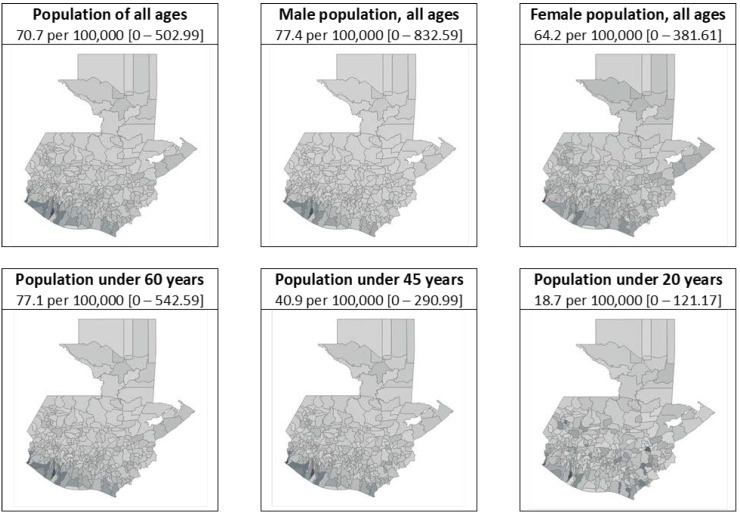
Municipal crude mortality rates for chronic kidney disease, Guatemala 2009–2019 (*N* = 340 municipalities), by gender and age group. Mean and range. Age groups are not mutually exclusive. Darker shades represent higher rates.

**Figure 3 ijerph-20-05532-f003:**
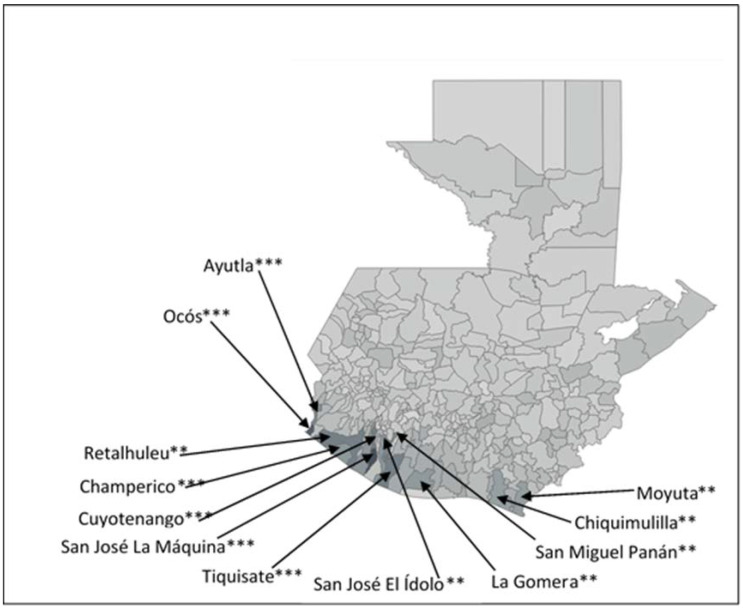
Municipalities that are part of the top 25 municipalities with highest CKD mortality rates all six age- or sex-specific groups ***, and in five age- or sex-specific groups **.

**Table 1 ijerph-20-05532-t001:** Number of deaths from chronic kidney disease, Guatemala 2009–2019 (*N* = 28,723).

	Male		Female		Total	
	Deaths	%	Deaths	%	Deaths	%
**By age group**						
Under 20 years	911	6%	879	7%	1790	6%
20–59 years	4968	33%	4239	31%	9207	32%
60+ years	9322	61%	8313	62%	17,635	61%
Not classified	58	0%	33	0%	91	0%
All ages	15,259	100%	13,464	100%	28,723	100%
**By ethnicity**						
Maya	2289	15%	2429	18%	4718	16%
Garífuna	6	0%	6	0%	12	0%
Xinka	4	0%	6	0%	10	0%
Mestizo/Ladino	7101	47%	5893	44%	12,994	45%
Other	126	1%	102	1%	228	1%
Ignored	1277	8%	1069	8%	2346	8%
Not classified	4456	29%	3959	29%	8415	29%
All ethnicities	15,259	100%	13,464	100%	28,723	100%

**Table 2 ijerph-20-05532-t002:** Descriptive statistics for municipal crude mortality rates per 100,000 for chronic kidney disease, by gender and age group *, Guatemala 2009–2019 (*N* = 340 municipalities).

	Population of All Ages	Male Population, All Ages	Female Population, All Ages	Population under 60 Years	Population under 45 Years	Population under 20 Years
Mean	70.66	77.38	64.22	77.08	40.89	18.72
Median	55.71	53.70	55.79	61.36	32.41	14.52
Standard Deviation	59.22	88.15	42.63	64.48	35.55	19.09
Minimum	0	0	0	0	0	0
Maximum	502.99	832.59	381.61	542.59	290.99	121.17
3rd Quartile	84.78	86.18	84.72	92.42	50.36	25.43
1st Quartile	37.73	32.61	36.29	40.64	21.47	5.51

*Age groups are not mutually exclusive.

**Table 3 ijerph-20-05532-t003:** Top-15 municipal crude mortality rates for chronic kidney disease, by gender and age group, Guatemala 2009–2019 *.

Municipality	Population of All Ages		Male Population, All Ages		Female Population, All Ages		Population under 60 Years		Population under 45 Years		Population under 20 Years	
	Rank	Mortality Rate	Rank	Mortality Rate	Rank	Mortality Rate	Rank	Mortality Rate	Rank	Mortality Rate	Rank	Mortality Rate
San José La Máquina ***	1	503.0	1	832.6	5	193.4	1	542.6	2	290.7	1	121.2
Ocós ***	2	452.0	3	524.5	1	381.6	2	488.7	1	291.0	2	102.9
Cuyotenango ***	3	332.0	2	537.7	18	134.7	3	360.7	6	172.1	15	60.9
Tiquisate ***	4	322.9	7	408.1	2	239.0	5	352.9	5	187.5	11	65.1
Retalhuleu **	5	322.6	4	499.3	9	156.5	4	354.0	3	200.5	41	39.2
San José El Ídolo **	6	303.6	5	422.0	6	191.0	6	330.7	9	132.0	34	41.9
Ayutla ***	7	280.7	10	350.2	3	215.1	7	302.2	4	190.6	10	67.1
San Miguel Panán **	8	271.3	6	414.4	20	133.3	8	290.1	13	114.3	45	37.4
San Andrés Villa Seca	9	240.5	8	382.9	56	99.8	9	258.6	11	130.2	64	30.0
Santo Domingo Suchitepéquez	10	236.5	9	368.0	38	110.9	10	256.1	12	129.3	166	15.0
Champerico ***	11	231.6	11	322.6	13	143.7	11	254.0	7	167.1	19	53.7
La Gomera **	12	225.0	12	315.0	19	133.8	12	243.2	14	104.4	215	9.9
Moyuta **	13	208.6	13	272.8	12	147.4	13	230.6	10	130.4	31	44.6
Chiquimulilla **	14	204.7	17	222.8	7	187.2	14	228.2	17	94.0	106	22.4
San Juan Bautista	15	166.1	16	232.6	54	101.1	15	178.0	108	45.5	267	0.0

* Mortality rates per 100,000 for the 2009–2019 period. ** Municipalities that are part of the top-25 municipalities with highest CKD mortality rates in all but one of the six age- or sex-specific groups. *** Municipalities that are part of the top-25 municipalities with highest CKD mortality rates in all six age- or sex-specific groups. Table with all 340 municipalities available in Appendix A.

**Table 4 ijerph-20-05532-t004:** Bivariate linear regression coefficients for agrarian territories and municipal mortality rates for chronic kidney disease, by gender and age group, Guatemala 2009–2019 (*N* = 28,723).

Agrarian Territories	Population of All Ages	Male Population, All Ages	Female Population, All Ages	Population under 60 Years	Population under 45 Years	Population under 20 Years
1. Northwest Highlands	−26.64 ***	−36.79 ***	−16.88 ***	−29.99 ***	−11.98 **	−3.22
2. North Transversal Strip	−42.22 ***	−52.28 ***	−32.55 ***	−47.01 ***	−21.11 ***	−7.55
3. Scattered in the north, east and south	12.43	14.80	10.34	13.40	3.23	−3.71
4. Concentrated in the south and dispersed in the highlands	35.96 ***	47.37 ***	24.71 ***	39.64 ***	23.06 ***	5.40
5. East	− 8.10	−15.00	− 1.38	− 7.073	−3.57	5.31 **
6. Western highlands, urban, food insecure	−11.29	−24.73	1.33	− 12.85	−8.01	−1.70
7. Southwest	51.09 ***	81.16 ***	22.20 ***	55.15 ***	24.15 ***	1.67
8. Petén	−43.81 **	−56.09 **	−31.71 **	−48.43 **	−22.78 **	−9.71
9. Urban and peri-urban	−7.66	−13.76	−1.66	−8.13	−8.03	−1.44
10. Metropolitan	26.36	30.23	23.19	31.79	14.64	15.14

*** *p* < 0.01, ** *p* < 0.05.

**Table 5 ijerph-20-05532-t005:** Bivariate linear regression coefficients for other relevant social and environmental indicators and municipal mortality rates for chronic kidney disease, Guatemala 2009–2019 (*N* = 28,723).

Social and Environmental Indicators	Population of All Ages	Male population, All Ages	Female Population, All Ages	Population under 60 Years	Population under 45 Years	Population under 20 Years
Average temperature	4.12 ***	5.90 ***	2.38 ***	4.52 ***	2.03 ***	0.43 *
Altitude	−0.02 ***	−0.03 ***	−0.01 ***	−0.03 ***	−0.01 ***	−0.01 *
Poverty	4.97 ***	5.88 **	4.10 ***	5.69 ***	2.17 **	0.94 **
Agriculture workforce	−1.75	−1.60	−1.92	−2.02	−0.19	−0.18
Agricultural use area	1.85	−0.81	4.40 *	1.99	−0.35	−0.72
Healthcare facilities: first level	−0.17	0.24	−0.57	−0.16	−0.12	−0.07
Healthcare facilities: total	0.04	0.40	−0.30	0.08	0.03	0.01
Mayan population	−38.63 ***	−52.63 ***	−25.17 ***	−43.62 ***	−20.30 ***	−6.90 **
Ladino/Mestizo population	39.87 ***	55.77 ***	24.55 ***	45.01 ***	20.83 ***	6.23 **

*** *p* < 0.01, ** *p* < 0.05, * *p* < 0.1.

**Table 6 ijerph-20-05532-t006:** Multivariate linear regression coefficients for significant social and environmental indicators and municipal mortality rates for chronic kidney disease, Guatemala 2009–2019 (*N* = 28,723).

Social and Environmental Indicators	Population of All Ages	Male Population, All Ages	Female Population, All Ages	Population under 60 Years	Population under 45 Years	Population under 20 Years
Agrarian territory 7: southwest	42.93 ***	65.32 ***	21.24 ***	46.09 ***	19.42 ***	−1.11
Agrarian territory 4: Concentrated in the south	51.69 ***	68.68 ***	35.09 ***	56.35 ***	29.93 ***	5.71 *
Average temperature	2.50	2.53	2.37	2.76	1.20	0.88
Altitude	−0.00	−0.01	0.00	−0.00	−0.00	0.00
Poverty	6.77 ***	8.16 **	5.43 ***	7.49 ***	2.90 **	0.77
Ladino/Mestizo population	−17.70 *	−23.02	−12.51	−17.66	−6.65	1.33

*** *p* < 0.01, ** *p* < 0.05, * *p* < 0.1.

## Data Availability

The data presented in this study are available in Appendix A: Database (spreadsheet).

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
