# Peer review of "Environmental and Social Factors Associated with High Chronic Kidney Disease Mortality Rates in Municipalities of Guatemala: An Ecological Study of Municipal-Level Mortality Data"

_ijerph, 2023, doi:10.3390/ijerph20085532_

Round 1
Reviewer 1 Report
In this nicely written paper, the author has added incrementally to our understanding of mortality rates related to the diagnosis of chronic kidney disease (CKD) in Guatemala using an ecological population-based design, scaled to the municipality level. It is consistent with earlier observations that there are higher rates of CKD (i.e. patients in dialysis) in agricultural regions of Guatemala, a country that has previously been reported to have rates of CKD and CKDnt. The paper is an incremental contribution to the existing literature, reinforcing the need to address CKD mortality in this and other Latin American countries.
The methods are solid and the interpretation of the data is appropriate. The use of the 2018 municipal population data as the sole denominator is unfortunate, but understandable and probably does not impact the general conclusions to be drawn from the paper, however, this limitation should be explored more in the discussion. Another limitation to be discussed is the usual concern with clinically-determined diagnosis codes.
Limitations of the data set are appropriately acknowledged by the author.
The paragraph starting on line 300 in the discussion, while very carefully worded with appropriate caveats, still overstates the relationship between the data and CKDnT.
Both in the introduction and in the discussion the author should reduce the amount written about CKDnT. Given the line of evidence that the majority of CKD mortality is related to diabetes and hypertension, arguably as much or more could have been written about those conditions. The data set concerns all CKD and is unable to offer specific insights into mortality rates attributable to CKDnT-related risks or to comorbidities such as diabetes, hypertension, or other causes. The paper teaches us about CKD mortality broadly, so I would keep the focus on CKD.
The effort to relate the data set to particular environmental factors like altitude and temperature is appreciated and defensible based on previous studies. However the use of average temperature is itself a complex variable: Average temperature is a very crude reflection of heat exposure (e.g. temp range, peak, duration, change over the decade, individual factors like access to cooling, shade, and other mitigations, migration patterns that influence individual exposure, are just a few of the factors that limit our ability to interpret the relationship of avg. temperature to mortality). This should be acknowledged in the discussion so that readers don't prematurely conclude that the absence of a relationship of avg temp with mortality means no effect.
I would encourage the author to add to the discussion their thoughts about the kinds of information and methods (e.g. additional data they might have liked to have had) that could be used in the future to improve our ability to look at the attribution of CKD mortality. What would be the author's call to action for ways of improving public health surveillance tools and analytic approaches with an eye toward improving national health systems' abilities to understand changes in CKD mortality rates, to address causes, and reduce mortality rates?
I congratulate the author on this thoughtful contribution to our understanding of CKD mortality in Guatemala and for continuing to draw attention to this devastating global problem.
Author Response
Author’s response to Reviewers 1 comments and suggestions
I thank the reviewer for the time spent reading the manuscript, and for the helpful feedback I received, which helps make this a stronger manuscript. I have copied the reviewer’s comments and suggestions in italics, where I also underlined the specific suggestions made by the reviewer. I have inserted my responses immediately after each of the paragraphs, with bullet points.
REVIEWER 1
Paragraph 1. In this nicely written paper, the author has added incrementally to our understanding of mortality rates related to the diagnosis of chronic kidney disease (CKD) in Guatemala using an ecological population-based design, scaled to the municipality level. It is consistent with earlier observations that there are higher rates of CKD (i.e. patients in dialysis) in agricultural regions of Guatemala, a country that has previously been reported to have rates of CKD and CKDnt. The paper is an incremental contribution to the existing literature, reinforcing the need to address CKD mortality in this and other Latin American countries.
- Author’s response: I thank the reviewer for this summary and general assessment of the manuscript.
Paragraph 2. The methods are solid and the interpretation of the data is appropriate. The use of the 2018 municipal population data as the sole denominator is unfortunate, but understandable and probably does not impact the general conclusions to be drawn from the paper, however, this limitation should be explored more in the discussion. Another limitation to be discussed is the usual concern with clinically-determined diagnosis codes.
- Author’s response: I added the following on lines 323-328 of the discussion:
- Another limitation is that, given that there are no available reliable population estimates by municipality for the intercensal years between 2002 and 2018, this analysis used the 2018 census data for the denominators; the effect of this limitation likely does not impact the general conclusions of this analysis, but it could have an effect on the mortality rates for specific municipalities that have had dramatic changes in fertility, mortality, or mi-gration rates, but the demographic information that would allow for the identification of those municipalities is not yet available.
- Author’s response: I believe that the limitation related to clinically-determined diagnosis codes is already addressed in the discussion, lines 346-352, which I copied here for your convenience:
- It is important to take into account that the analyzes that come from vital statistics databases anywhere in the world have limitations in the quality of the definition of the variables and in the consistency with which they are interpreted by those who generate the data. The strength of these data is that they cover the entire population and are generated periodically. It is worth considering that the quality of the data generated by the Guatemalan vital statistics system has recently been assessed as good (66), and as “four stars” out of five (1).
Paragraph 3. Limitations of the data set are appropriately acknowledged by the author.
- Author’s response: I take this comment as being consistent with my responses to the reviewer’s comments on paragraph 1. I made no further changes based on this.
Paragraph 4. The paragraph starting on line 300 in the discussion, while very carefully worded with appropriate caveats, still overstates the relationship between the data and CKDnT.
- Author’s response: I changed the paragraph according to the reviewer’s comment, and I pasted it here (starting now on line 311):
- In terms of the public health and epidemiological investigation of traditional and novel or emerging risk factors and determinants of CKD, this study’s findings seem to support hypotheses that point to environmental factors related to agricultural use of land, and to poverty. Although it is reasonable to interpret from these findings that there are traditional risk factors like diabetes and hypertension driving most of the CKD deaths, it also seems plausible to identify a cluster of municipalities with very high mortality rates across different age- and sex-specific groups where some type of environmental exposure associated to agricultural use of land and/or to poverty is also at play in tandem with traditional risk factors of CKD. Given that exposure to agrochemicals or heavy metals has been hypothesized to play a role in CKDnT but that available evidence is inconclusive (4-13), the environmental study of the identified cluster of municipalities offers a promising way forward for testing environmental hypotheses. Similarly, it is known that poverty is associated to traditional risk factors like diabetes and hypertension, but it may be associated to other factors that affect people at younger ages, such as nutrition, as some studies have found (65), or to other cardiometabolic exposures.
Paragraph 5. Both in the introduction and in the discussion the author should reduce the amount written about CKDnT. Given the line of evidence that the majority of CKD mortality is related to diabetes and hypertension, arguably as much or more could have been written about those conditions. The data set concerns all CKD and is unable to offer specific insights into mortality rates attributable to CKDnT-related risks or to comorbidities such as diabetes, hypertension, or other causes. The paper teaches us about CKD mortality broadly, so I would keep the focus on CKD.
- Author’s response: I agree with the reviewer’s suggestion. I significantly reduced the first paragraph of the introduction (pasted below, starts on line 28 of the manuscript), and to the second-to-last paragraph of the discussion (pasted above, in response to the reviewer’s comments on paragraph 3), which were the only ones that focused directly on CKDnT, although I kept some references to CKDnT because I cannot completely ignore them in light of the results.
- The incidence, prevalence, and mortality from chronic kidney disease (CKD) has increased globally in the past four decades (1). The worldwide rise in CKD cases is attributable to the increased number of people living with diabetes and high blood pressure, with other, less common causes such as infection, injury, exposure to toxins, or environmental pollution on the rise in some world regions (1). A type of CKD that is not related to diabetes and hypertension has been described as CKD of non-traditional causes (CKDnT) (2-7). Although this paper focuses on mortality from CKD in general, it remains aware that CKDnT has reached epidemic proportions in Central America, making it a public health priority (4), and that studies that focus on CKDnT may shed some light on the present study’s findings. Most researchers attribute CKDnT to multifactorial causes, with an interplay of repeated heat exposure and poor hydration working in synergy with exposure to toxins and infectious agents to damage the kidneys (2–6). While environmental factors such as exposure to agrochemicals or heavy metals have been hypothesized to play a role in CKDnT, available evidence is inconclusive (4-13), and research on this area is insufficient in Latin American countries (10).
Paragraph 6. The effort to relate the data set to particular environmental factors like altitude and temperature is appreciated and defensible based on previous studies. However the use of average temperature is itself a complex variable: Average temperature is a very crude reflection of heat exposure (e.g. temp range, peak, duration, change over the decade, individual factors like access to cooling, shade, and other mitigations, migration patterns that influence individual exposure, are just a few of the factors that limit our ability to interpret the relationship of avg. temperature to mortality). This should be acknowledged in the discussion so that readers don't prematurely conclude that the absence of a relationship of avg temp with mortality means no effect.
- Author’s response: I agree with the reviewer’s comment, and added the following sentences on lines 257-261 of the discussion
- The lack of association with temperature, altitude, and ethnicity should not be inter-preted as a lack of relevance of these variables in the etiology of CKD, but as directly related to the ecologic nature of this analysis, especially since these variables interact dynamically with other variables at the individual level of analysis, which was outside of the scope of the present study.
Paragraph 7. I would encourage the author to add to the discussion their thoughts about the kinds of information and methods (e.g. additional data they might have liked to have had) that could be used in the future to improve our ability to look at the attribution of CKD mortality. What would be the author's call to action for ways of improving public health surveillance tools and analytic approaches with an eye toward improving national health systems' abilities to understand changes in CKD mortality rates, to address causes, and reduce mortality rates?
- Author’s response: I appreciate this encouragement from the reviewer. I have expanded the second paragraph of the conclusion (lines 360-369), which now reads:
- Future research aimed at investigating social and environmental exposures in high CKD mortality municipalities is needed in addition to population-based studies using a life-cycle approach, including gender differences, and the role of occupational trajectories. Ideally, an interdisciplinary, multi-level cohort study that collects data at the individual and community levels, and that also collects social and environmental data would help get a more comprehensive view of the drivers of increased mortality rates in these municipalities. Public health interventions aimed at early detection of decreasing renal function in high CKD mortality municipalities are recommended. Public health interventions should begin with active surveillance using screening tests from individuals’ renal function, as well as screening tests from potential community sources of pollutants.
Paragraph 8. I congratulate the author on this thoughtful contribution to our understanding of CKD mortality in Guatemala and for continuing to draw attention to this devastating global problem.
- Author’s response: I appreciate the reviewer’s encouraging words.

Reviewer 2 Report
Please clarify what you mean by the sentence on line 105 "The study is an ecological, cross-sectional analysis...".
What study design was actually used in this study? Ecology and cross-sectional are two different study designs, why did you include these two designs in your research?
Your statistical analysis is on aggregate data, not individual data. Therefore, the term you use regarding "confounding variables" (line 223 & 234) is inappropriate. If you consider that temperature, poverty, and ethnicity are confounding variables, then what variables is the main risk factor in your research?
Figure 2 is very uninformative and difficult to understand.
You conclude that environmental factors play a role in the high mortality associated with CKD. However, I could not find a link between these conclusions from the results of the research that you presented.
Author Response
Author’s response to Reviewer 2 comments and suggestions
I thank the reviewer for the time spent reading the manuscript, and for the helpful feedback I received, which helps make this a stronger manuscript. I have copied the reviewer’s comments and suggestions in italics, where I also underlined the specific suggestions made by the reviewer. I have inserted my responses immediately after each of the paragraphs, with bullet points.
ENGLISH LANGUAGE REVIEW:
Given that Reviewer 2 marked the article as needing to undergo extensive English revisions, and even if Reviewer 1 marked it as having an adequate level of English, I wanted to include this response for your consideration:
- The article had been reviewed by two English native speaker colleagues prior to my initial submission.
- The revised version has been reviewed by an English native speaker, writing consultant at the University of Denver Writing Center, who made minor suggestions I have already incorporated in the revised version I am submitting.
REVIEWER 2
Paragraph 1. Please clarify what you mean by the sentence on line 105 "The study is an ecological, cross-sectional analysis...".
- Author’s response: I realize that the reviewer is right about the potential confusion generated by my choice of words, given that the term “cross-sectional” can have two meanings when discussing public health and epidemiology research. One meaning refers to the distinction between a “longitudinal” versus a “cross-sectional” approach; and the other one refers to the “cross-sectional study” also known as prevalence study, and transversal study. Colleagues who have read previous drafts have suggested to clarify that the study is “cross-sectional,” meaning that it is not longitudinal, given that I am using data from 2009 to 2019. I have changed my word choice in the title and abstract, where I now say simply that this is an ecological study. I have also changed the paragraph that now starts on line 108 of the methods section, which I pasted here:
- This is an ecological study of municipal-level social and environmental factors associated with chronic kidney disease (CKD) mortality in Guatemala from 2009-2019, doing a cross-sectional (i.e., not longitudinal) analysis, and using vital registration data.
Paragraph 2. What study design was actually used in this study? Ecology and cross-sectional are two different study designs, why did you include these two designs in your research?
- Author’s response: It is an ecological study. I explain in my response to paragraph 1 of this reviewer’s comments the reason why I had included the term cross-sectional, and the changes I made in the title, abstract, and methods section.
Paragraph 3. Your statistical analysis is on aggregate data, not individual data. Therefore, the term you use regarding "confounding variables" (line 223 & 234) is inappropriate. If you consider that temperature, poverty, and ethnicity are confounding variables, then what variables is the main risk factor in your research?
- Author’s response: I agree with the reviewer’s comment that the term “confounding variables” is misleading and inappropriate. I have eliminated the term on lines 147, 227, and 238.
Paragraph 4. Figure 2 is very uninformative and difficult to understand.
- Author’s response: I appreciate this comment from the reviewer. Colleagues who read earlier versions of this paper suggested me to include this figure, as it would help readers who are unfamiliar with the geography of Guatemala, by complementing the information provided in tables. I have kept the figure because it may help at least some readers, but if the reviewers and/or editor feel strongly about the need to eliminate it, I will eliminate it. I added the following sentence on lines 182-183, just before the figure is presented:
- Figure 2 shows this geographic distribution for the total population, and for each of the age and sex groups, with the darkest shaded municipalities representing the highest mortality rates.
Paragraph 5. You conclude that environmental factors play a role in the high mortality associated with CKD. However, I could not find a link between these conclusions from the results of the research that you presented.
- Author’s response: I appreciate this comment from the reviewer. I eliminated some sentences from the conclusions where I mentioned “environmental factors” in general, since they may have introduced confusion to my conclusions. I have only kept the sentence where I mention “environmental factors related to agricultural use of land,” which is directly consistent with what figure 1, table 4, table 5, and table 6 show. The shorter conclusion paragraph on lines 353-355 now reads:
- The results from this study suggest that social factors related to poverty, and environmental factors related to agricultural use of land play a role in the high CKD mortality rates documented in a cluster of municipalities of Guatemala.